# Expression of Autophagic and Inflammatory Markers in Normal Mucosa of Individuals with Colorectal Adenomas: A Cross Sectional Study among Italian Outpatients Undergoing Colonoscopy

**DOI:** 10.3390/ijms23095211

**Published:** 2022-05-06

**Authors:** Paola Sena, Stefano Mancini, Monica Pedroni, Luca Reggiani Bonetti, Gianluca Carnevale, Luca Roncucci

**Affiliations:** 1Department of Surgery, Medicine, Dentistry and Morphological Sciences with Interest in Transplant, Oncology and Regenerative Medicine, University of Modena and Reggio Emilia, Via del Pozzo 71, 41124 Modena, Italy; gianluca.carnevale@unimore.it; 2Department of Internal Medicine and Rehabilitation, Santa Maria Bianca Hospital, Mirandola 6, 41037 Modena, Italy; stefano.mancini@ausl.re.it; 3Department of Medical and Surgical Sciences, University of Modena and Reggio Emilia, Via del Pozzo 71, 41124 Modena, Italy; monica.pedroni@unimore.it (M.P.); luca.reggianibonetti@unimore.it (L.R.B.); luca.roncucci@unimore.it (L.R.)

**Keywords:** adenoma, colorectal mucosa, autophagy, inflammation, cross sectional study

## Abstract

Colorectal cancer (CRC) ranks among the three most common cancers in terms of both cancer incidence and cancer-related deaths in Western industrialized countries. Lifetime risk of colorectal cancer may reach 6% of the population living in developed countries. In the current era of personalized medicine, CRC is no longer considered as a single entity. In more recent years many studies have described the distinct differences in epidemiology, pathogenesis, genetic and epigenetic alterations, molecular pathways and outcome depending on the anatomical site. The aim of our study is to assess in a multidimensional model the association between metabolic status and inflammatory and autophagic changes in the normal colorectal mucosa classified as right-sided, left-sided and rectum, and the presence of adenomas. One hundred and sixteen patients undergoing colonoscopy were recruited and underwent a complete serum lipid profile, immunofluorescence analysis of colonic biopsies for MAPLC3 and myeloperoxidase expression, matched with clinical and anthropometric characteristics. Presence of adenomas correlated with cholesterol (total and LDL) levels, IL-6 levels, and MAPLC3 tissue expression, especially in the right colon. In conclusion, serum IL-6 amount and autophagic markers could be good predictors of the presence of colorectal adenomas.

## 1. Introduction

Colorectal cancer (CRC), despite ranking third among the most commonly diagnosed cancers worldwide [1], is one of the most preventable cancers because it usually arises in benign lesions, i.e., adenomas, which evolve into CRC over time. The slow polyp-cancer progression sequence seen in the general population offers an opportunity to detect and remove polyps before they undergo malignant transformation [2]. CRC screening guidelines recommend testing asymptomatic subjects of both sexes aged older than 50 years for either fecal occult blood test or endoscopic examinations, such as sigmoidoscopy or colonoscopy [3,4]. For high-risk patients who have a family history of CRC in first-degree relatives, colonoscopy is the recommended screening tool [5]. However, evidence is growing about the relative impact of different factors in CRC development and in the onset of adenomatous polyps [6]. Stratifying the population into risk categories in accordance with a precision medicine guideline offers the potential to improve the efficiency of CRC screening by tailoring the intensity of the preventive approaches to the predicted level of risk. This is particularly true for the risk of sporadic CRC in the aged population (76 to 85 years), for which the decision to screen for colorectal cancer should be an individual one, considering the patient’s overall health and prior screening history [7].

In 2016, Usher-Smith JA et al. identified eighty-seven risk factors (excluding genes and SNPs) implicated in CRC pathogenesis and thus possibly involved in different models of CRC risk score [8]. Among these, age, gender, and BMI are emerging as reliable predictors to develop CRC [9]. The increasing evidence of a strong association between obesity, male sex, advanced age (and also physical inactivity as well as tobacco use) [10], with a higher risk of CRC has driven the attention of the researchers to investigate the common pathways and clinical features shared between colorectal cancer risk and cardiovascular risk. Most CRC is sporadic and largely attributable to the presence of environmental risk factors that characterize westernization (e.g., high BMI, physical inactivity, poor diets, alcohol consumption and smoking) which can be modifiable. This fact is evident as the burden of the CRC is shifting to low and middle-income countries due to their westernization [10]. Another significant finding is that the increase in the incidence of CRC at a younger age (before age 50) is an emerging trend [11]. Nowadays, it is an accepted concept that the presence of a subclinical inflammatory status is another risk factor for developing cancer (not only colorectal) [12], and serum lipid profile seems to have a close association with colorectal cancer, both in pre-clinical and clinical studies [13]. Moreover, the infiltration of colonic mucosa by neutrophils and macrophages may promote carcinogenesis through myeloperoxidase (MPO), a key enzyme contained in the lysosomes of neutrophils that regulates local inflammation and the generation of reactive oxygen species (ROS) and mutagenic species [14]. However, areas of uncertainty still remain on the role of systemic inflammatory markers, lipid profile, and the occurrence of adenomas at colonoscopy in substantially healthy people. It must be emphasized that during the adenoma–carcinoma sequence, key regulating proteins of appropriate mucosal cell death undergo changes in expression. Normal cells evolve progressively to a neoplastic state, acquiring a series of hallmarks or biological capabilities, among which the ability to resist cell death is of considerable importance [2]. In the multistep process of human carcinogenesis, cells accumulate multiple genetic alterations as they progress to a more malignant phenotype. Apoptosis and autophagy are regulated by common upstream signals, and share common components and functional relationships. As far as cancer development is concerned, the two process commonly occur in the same cells; in some cases, inhibition of apoptosis causes autophagy, while, in other cases, inhibition of autophagy triggers apoptosis [15]. During the early steps of cancer development, autophagy may have a suppressive effect, whereas it may act alternatively as a tumor promoter during later stages of carcinogenesis.

To our knowledge, no study has aimed at evaluating the connection between the autophagic/inflammatory status of the normal mucosa in patients with colorectal adenomas with respect to patients with a negative colonoscopy. On this premise, we aimed at investigating the correlation between the expression of autophagic/inflammatory markers in the normal colorectal mucosa, the pattern of the patient’s serum lipid profile, and systemic inflammation in relation to the presence of adenomatous polyps at colonoscopy.

## 2. Results

### 2.1. Patterns of Myeloperoxidase (MPO) Expression in Normal Mucosa Samples (Right Colon, Left Colon and Rectum)

In order to evaluate the expression pattern of the MPO protein, as a marker of inflammatory infiltration of the mucosa, in samples of normal mucosa of the right colon (NMRC, cecum or ascending colon), left colon (NMLC, descending colon) and rectum (NMR), immunofluorescence experiments coupled with confocal analysis were performed. This technique is an excellent morphological tool that can be used to quantify MPO immunoreactivity levels and its cellular location, in very thick samples. The staining patterns of MPO appeared coherent and unchanged in the different optical sections of the same sample, without appreciable variations in intensity and cellular localization. In all NMRC samples, a broad shaggy intercellular stromal fluorescence is evident, indicating an intense MPO reactivity. As expected, epithelial cells were not stained (Figure 1A). All NMLC specimens showed a moderate MPO expression, while in NMR the staining drastically decreases (Figure 1B,C).

The semiquantitative evaluation of immunostaining intensity, reported as Immunofluorescence Intensity Score (IFIS) in Table 1, showed that the level of MPO protein in the stromal compartments had a decreasing trend from the right colon to the left colon to the rectum, and this trend was statistically significant (Table 1).

### 2.2. MAP-LC3 Autophagic Activity in Normal Mucosa Samples (Right Colon, Left Colon, and Rectum)

To ascertain the presence of autophagy in normal mucosa samples, immunostaining of the Microtubule-associated protein light chain 3 (MAP-LC3) using fluorescent antibodies against MAP-LC3 was carried out. MAP-LC3 is known as an autophagosome specific marker, and the puncta formation of MAP-LC3 in the autophagosome can be carefully detected by confocal microscopy, which is widely used as a reliable method for characterize autophagy [16].

Two distinct autophagic features were evident in MAP-LC3 stained positive cells: a diffuse and granular staining dispersed in the cytoplasm (Figure 1G), or a rounded and coarse stained material enclosed within a cytoplasmic vacuole that accumulates around the nucleus (Figure 1H). Dense and rounded autophagosomes, variable in size and density, were clearly evident in stromal cells (Figure 1D–F). Both patterns of expression, often similar to small clumps, were noted in many stromal cells of all sections examined, although to different extents and labelling intensity in the right colon, left colon, and rectum (Figure 1D–F).

Regarding the semiquantitative analysis of immunofluorescence intensity, there were substantial differences between NMRC and NMLC; NMR showed a very low immunoreactivity, hardly detectable in many samples. Indeed, the IFIS decreased sharply from the right colon to the rectum (Table 2).

### 2.3. Demographic, Anthropometric, and Clinical Features of Patients According to the Presence of Colorectal Adenomas

Demographic, anthropometric, and clinical characteristics of patients are summarized in Table 3.

One hundred and sixteen patients were recruited (M/F 64/52) with a mean age of 60.8 years (M/F, 60.8/60.9). Fifty one out of 116 were positive for at least one adenoma at colonoscopy.

As reported in Table 3, adenomas were significantly more frequent in men, in fact, of the 51 colonoscopy positive patients, 37 were men and 14 women (*p* < 0.001).

The individual mean weight in our sample was 77.8 Kg, with a mean body mass index (BMI) indicative for overweight (27.4 ± 4.6 Kg (m^−2^)) and, concordantly, increased mean values of waist circumference (99.5 ± 15.5 cm) and waist to hip ratio (0.97 ± 0.10). Moreover, as reported in Table 3, mean systolic blood pressure was higher than the normal recommended levels (146.7 ± 18.5 mmHg), whereas mean diastolic pressure was within the normal range (83.4 ± 10.4 mmHg). Table 3 also reports the mean values for some biochemical serological parameters, cigarette smoking, tobacco use and sedentary lifestyle. The overall occurrence of tobacco use was 37.5% (43 patients of 116), 67.2% (78 patients of 116 for low to moderate (3 drinks per week) intake of alcohol), and 42.2% (49 of 116 patients) for sedentary lifestyle (less than 30 min walk per day).

None of the lifestyle factors considered were significantly different in patients with or without adenomas at colonoscopy (Table 4).

### 2.4. IFIS for MAP-LC3 and MPO Protein Expression in Patients with Colorectal Adenomas

The expression of MAP-LC3, reported as IFIS, revealed a strong positive correlation with the presence of adenomas, especially in the right (cecum-ascending colon) and left (descending-sigmoid) colon, whereas no correlation was found between MAP-LC3 expression and the presence of adenomas in the rectum, as reported in Table 5.

On the contrary, the analysis of the logistic correlation between the values of the inflammatory infiltration marker, MPO and the presence of adenomas, did not reach a statistically significant result, in any of the three tracts of the colon (left, right and rectum).

### 2.5. IFIS for MAP-LC3 and MPO Protein Expression, and Clinical Characteristics

The evaluation of the correlations between the IFIS for MAP-LC3, in the different tracts of the colorectum, and the single clinical characteristics of patients was the following: there was a strongly positive correlation between MAP-LC3 expression in the mucosa of the right colon and serum IL6 values (*p* = 0.02) (Figure 2A); a strong positive correlation was also found between MAP-LC3 in the right colon and the values characterizing the lipid profile of the patients, i.e., LDL and total cholesterol (*p* = 0.003 for LDL and *p* = 0.006 for total cholesterol) (Figure 2B,C).

In the left colon, the expression of MAPLC showed a positive correlation only with serum IL-6 (*p* = 0.002) (Figure 2D) and a weak positive correlation with LDL (*p* = 0.055). On the other hand, the analysis of the correlation between the expression of MAPLC3 in the rectum and the different patient’s clinical parameters was not significant. Quite unexpectedly, the quantification of MPO expression in the three tracts of the colon did not show significant correlations with any of the parameters examined.

### 2.6. Multivariate Analysis

When the variables significantly related to autophagy in the right colonic mucosa in univariate analyses were evaluated with a multiple regression model, the serum levels of Il-6 were the best predictors of MAP-LC3 expression. The presence of at least one adenoma at colonoscopy was also independently associated with autophagy in the right colonic mucosa (Table 6).

## 3. Discussion

This study highlighted, for the first time, the interplay between the expression of autophagic mediators, the presence of tissue and systemic inflammation, and specific clinical parameters in patients undergoing colonoscopy. These data, already interesting in themselves, are further enforced by the observation that there is a different expression of these markers in the various tracts of the large bowel, i.e., right colon, left colon, and rectum. In fact, our findings showed that there is a drastic, statistically significant, decrease in the expression pattern of both autophagic and inflammatory markers proceeding from the right colon to the rectum.

Furthermore, dividing the patients according to the presence of adenomas during colonoscopy, a relevant correlation was found between the high expression of the autophagic marker, MAPLC3, in the right colon and the occurrence of adenomas. This correlation held true also in the left colon, where the values of MAPLC3 appeared decreased but still consistent, whereas they were lost in the rectum, where the autophagic activity was almost completely undetectable.

The controversial role of autophagy in CRC development has been supported by a plethora of data [17,18]. Before tumorigenesis, autophagy can suppress tumor initiation by protecting normal cells and inhibiting inflammation (including inhibiting the inflammasome and necrosis) [19]. In contrast, autophagy promotes tumor growth in established tumors. Cancer cells have been found to require high baseline levels of autophagy for cell proliferation and advanced cancers appear to be dependent on autophagy to maintain their energy balance [20].

While autophagy has been extensively studied in colorectal cancer, where its significance is still unclear, only a few authors have attempted to explain its role in the normal mucosa of patients [21,22].

In fact the role of autophagy in cancer is controversial, but it was reported to be upregulated in colon cancer as evaluated by an increase of autophagic mediator expression in a significant proportion of primary tumors as compared with normal adjacent tissue [23,24]. In a previous study [25], we evaluated the involvement of autophagic activity in colorectal carcinogenesis by analyzing its regulation in preneoplastic lesions and in the tumor, according to the presence or absence of DNA microsatellite instability. Moreover, we demonstrated that autophagy was upregulated in carcinomas classified as DNA microsatellite stable (MSS) with respect to unstable ones (MSI). The importance of genetic stabilization by the DNA mismatch-repair (MMR) system is illustrated by the fact that defects in the human MMR pathway may contribute to the development of 15–20% of human sporadic CRCs [26], and MSI tumors have a better stage-adjusted survival compared with microsatellite stable tumors [27]. Our data suggested that autophagy is activated in human precancerous and cancerous cells and may represent a pro-survival mechanism for tumour cells. In recent years, much attention has been paid to the site of primary cancer in the treatment of colorectal cancer, it seems that the distinction between right colon cancer (RCC), left colon cancer (LCC) and cancer of the rectum may have a role as a prognostic factor. Many studies have pointed out that such a distinction is necessary due to their different outcomes, prognosis and clinical responses to chemotherapy [28]. Several studies confirmed that right-sided colon cancer has a worse prognosis than left-sided colon and rectal cancer [29,30,31]. The clinical and experimental data reported above could explain the higher autophagic activity found in the right colonic mucosa, compared to the other two tracts. This observation is also supported by the relevant correlation between high values of the autophagic marker and the presence of adenomas. Therefore, it is possible to hypothesize that the autophagic process may be an excellent predictor of the presence of adenomas.

Unexpectedly, the rate of neutrophilic/macrophage infiltration, tested with the quantification of myeloperoxidase expression, which showed a decreasing trend moving from the proximal colon to the rectum, did not reach any positive correlation with the presence of adenomas. Our collected reports pave the way for different interpretations: while on the one hand it is well known in the literature that inflammation plays a fundamental role in the onset of colorectal cancer [32], on the other hand MPO expression is a marker of non-specific immune activity able to recruit other immune cells [33]. Interestingly neutrophils are perhaps the least studied inflammatory cell type in CRC, although they are an important component of the inflammatory response. Important prognostic information can be obtained by analyzing the infiltration of specific subtypes of inflammatory cells (e.g., macrophages and lymphocytes) in colorectal cancer and implementation of an immunoscore in clinical practice is under study [34]. However, in colorectal tumorigenesis, the role of neutrophilic infiltration in patient prognosis has been the subject of conflicting reports, indicating both a positive and a negative effect [35,36,37,38]. This agrees with the absence of a positive statistic correlation between the increased presence of neutrophilic infiltrate in the right colon and the presence of adenomas found at colonoscopy. These data lead to the conclusion that the neutrophil/macrophagic infiltration rate may not be a good predictor of the presence of adenomas.

The correlation of autophagic marker, MAPLC3, and myeloperoxidase expression with different clinical characteristics, such as the systemic inflammatory state and the lipid profile of the patients was then evaluated.

Interestingly, a strongly positive correlation was found between the elevated expression of the autophagic marker in both the right and left colonic mucosa with serum levels of IL-6, indicating a systemic inflammatory state. Moreover in a multiple regression model, the only variables that were independently related to the expression of the autophagic protein were the presence of adenomas at colonoscopy and the serum levels of IL-6. In the rectum, the correlation was not statistically significant. The CRC diagnostic process is complex and involves the use of imaging procedures and laboratory markers. One of the most used biomarkers in the last decade is the carcinoembryonic antigen (CEA), which has the defect of not being useful in screening due to its low diagnostic sensitivity in the early stages of CRC [39,40]. Other serum biomarkers, including cancer antigen 19-9 (CA 19.9), tissue polypeptide specific antigen (TPS) and tumor-associated glycoprotein-72 (TAG-72), were studied without demonstrating acceptable diagnostic performance [41]. Łukaszewicz-Zając et al. in 2021 published a review in which they report a series of research papers exploring the clinical significance of selected inflammatory mediators as potential biomarkers [42].

Among these mediators, IL-6 stands out, as the authors showed that serum IL-6 is significantly higher in patients with CRC than in healthy subjects and suggested the usefulness of serum IL-6 measurements in CRC diagnosis. The correlation between the expression of a tissue marker of autophagy, the presence of adenomas and the serum levels of IL-6 can therefore be a starting point in the search for new diagnostic tools for the early diagnosis of CRC, to improve management and patient outcomes with CRC.

As for the correlation between MAPLC3 expression, the presence of adenomas and the patients’ total cholesterol values, there was a perfect agreement with the data reported in the literature; in fact clinical and experimental evidence supports the contention that lipid metabolism is involved in the development of colorectal cancer [43,44]. Several endoscopic studies on adenomas have investigated possible associations with serum concentrations of total cholesterol, high-density lipoprotein (HDL), low-density lipoprotein (LDL) or triglycerides. In a meta-analysis of studies from the United States, Europe and Asia, Tian et al. [45] reported that higher LDL, triglyceride and total cholesterol levels were associated with a higher prevalence of colorectal adenomas.

Other interesting findings are related to the analysis of the correlation between anthropometric measures, lifestyle and gender differences with the presence of adenomas at endoscopy. Although many reports show that excess body fat is a risk factor for colorectal cancer, the extent of the association of BMI and other indicators of overweight with long-term risk of CRC remains unclear [46]. The data collected did not show a statistically significant correlation between waist circumference, waist-to-hip ratio (used as an indicator of central adiposity), BMI and the presence of adenomas.

Gender differences in predisposition to colorectal cancer and the presence of adenomas have long been reported; although there are some discordant papers, most of the literature data show that men have a higher risk of developing colorectal polyps and cancers [47,48,49] and a meta-analysis also showed strong evidence that men have a higher risk of developing advanced colorectal cancer compared to women. Recently Ku et al. [50] showed, in a prospective observational study, that there is a higher incidence of adenomas in men than in women.

Our results are in line with the above reports as we found, stratifying by gender, the presence of adenomas in a statistically higher percentage of men than women. One of the most accepted hypotheses to explain this gender difference is that hormone replacement therapy in postmenopausal women is a preventative of colorectal cancer. Estrogens are associated, based on the ratio of their ERα/ERβ receptors, to anti-proliferation and apoptosis, important in the protection against colorectal cancer tumorigenesis [51,52].

The clinical implication of our findings may be viewed in the context of the definition of the individual colorectal cancer risk. Taking into account our previous works [53], it is possible to better define the individual risk measuring some anthropometric parameters, the serum IL-6 levels and the lipid profile, and the IFIS for the autophagic marker on a biopsy of normal mucosa in the right colon, along with the presence of adenomas at colonoscopy. Of course that would be only an estimate, but on clinical grounds, it could help in tailoring preventive measures to each individual.

In conclusion, the novelty of our research is represented by the intent to identify possible correlations between tissue biomarkers, which can represent good predictors of the presence of adenomas in the patient’s healthy mucosa, and serum factors that are analyzed in the normal laboratory routine. In the future, this approach could be useful in stratifying patients according to the risk of colorectal adenomas.

## 4. Materials and Methods

### 4.1. Study Population

This was a cross-sectional study, recruiting a total of 116 patients (M/F 64/52) with a mean age of 60.8 years (M/F 60.8/60.9), after written informed consent, undergoing colonoscopy for positive fecal blood test and/or abdominal symptoms, with a negative history for cancer or inflammatory bowel diseases. Patient enrollment started on 1st January 2015 and ended on 30th June 2021.

### 4.2. Statement of Ethics

The study was approved by the competent Ethic Committee (code no. 245/11) and the Local Health Agency of Modena. Every patient enrolled in the study signed a detailed written informed consent. The study was carried out according to the Declaration of Helsinki, to the Good Clinical Practice principles for medical research and to the current regulations relating to the protection and processing of personal and sensitive data (European Regulation *n*. 679/2016).

### 4.3. Data Collection

For each enrolled patient, anthropometric assessment (height, waist circumference, body weight, BMI (body weight in kg/height in m^2^), waist-to-hip ratio, and blood pressure), and clinical parameters (arterial pressure, a complete medical and drug history, a fasted serum glycemia) were taken. Patients with one or more adenomas, histologically confirmed, identified during colonoscopy, were considered positive, whereas patients with no adenomas were considered negative. Patients were studied in a blinded fashion for anthropometric data, other clinical records and the outcome of colonoscopy. Patients younger than 18 years, and with malignancy at any site were excluded.

### 4.4. Lipid Profile, C-Reactive Protein and IL-6

A fasting blood draw before the execution of colonoscopy was obtained in order to assess complete standard serum lipid profile, C-Reactive Protein (CRP) and Interleukin-6 (IL-6) levels. Lipid profile included total cholesterol, low-density lipoprotein cholesterol (LDL-cholesterol), high-density lipoprotein cholesterol (HDL-cholesterol), and triglycerides. Sera were immediately extracted by centrifugation and conserved at −80 °C until test execution.

Tests were performed on automated instrumentation Roche / Hitachi (Cobas 6000 Modular Analytics, F. Hoffmann-La Roche AG, Basel, Switzerland). Hs-CRP and IL-6 were measured by immunoturbidimetric technique with monoclonal antibodies bound to latex particles (dual-radius technology enhanced latex). Recommendable cut-off limits for the tests were 200 mg/dL or lower for total cholesterol, 120 mg/dL or lower for LDL-cholesterol, 40 mg/dL and 50 mg/dL or higher, for males and females respectively, and 150 mg/dL or lower for triglycerides. The lowest detectable limits for high-sensitive CPR (hsCRP) and IL-6 were 0.1 mg/L and 1.5 pg/mL, respectively.

### 4.5. Evaluation of Immunofluorescence by Confocal Microscopy on Colonic Biopsies

Three samples of normal mucosa (NM) were collected during colonoscopy (right colon (NMRC), left colon (NMLC), and rectum (NMR)). The samples of normal mucosa collected were fixed in 3% formalin and embedded in paraffin for immunofluorescence analysis as described above. Before immunofluorescence, routine histology of all tissue samples was carried out after H&E staining of the sections. Slides were dried overnight at 37 °C, dewaxed in two changes of fresh xylene, and rehydrated in a descending alcohol series. Antigen retrieval involved treatment with a protease (Pronase 1:20; Dako Cytomation) for 7 min. at 37 °C. After treatment with 3% BSA in PBS for 30 min. at room temperature, the paraffin sections were incubated with the primary antibodies (rabbit anti-human MAPLC3, Santa Cruz; mouse anti-human myeloperoxidase (MPO), abcam), diluted 1:25 in PBS containing 3% BSA for 1 h at room temperature. After washing in PBS, the samples were incubated for 1 h at room temperature with the secondary antibodies diluted 1:20 in PBS containing 3% BSA (sheep anti-mouse FITC conjugated, goat anti-rabbit TRITC conjugated, SIGMA). After washing in PBS and in H_2_O, the samples were counterstained with 1 mg/mL DAPI in H_2_O and then mounted with anti-fading medium (0.21 M DABCO and 90% glycerol in 0.02 M Tris, pH 8.0).

Negative control samples were not incubated with the primary antibody. The confocal imaging was performed on a Leica TCS SP2 AOBS confocal laser scanning microscope. Excitation and detection of the samples were carried out in sequential mode to avoid overlapping of signals. Sections were scanned with laser intensity, confocal aperture, with gain and black level setting kept constant for all samples. Optical sections were obtained at increments of 0.3 mm in the z-axis and were digitized with a scanning mode format of 512 × 512 or 1024 × 1024 pixels and 256 grey levels. The confocal serial sections were processed with the Leica LCS software to obtain three-dimensional projections. Image rendering was performed by Adobe Photoshop software. The original green fluorescent confocal images were converted to grey-scale and median filtering was performed. An intensity value ranging from 0 (black) to 255 (white) was assigned to each pixel. Background fluorescence was subtracted and immunofluorescence intensity (IF) was calculated as the average for each selected area. The fluorescence intensity at the selected areas, linearly correlated with the number of pixels, was quantitatively analyzed using the standard imaging analysis software of a NIS-Elements System. To each sample was assigned a code number and the score, referred to as ImmunoFluorescence Intensity Score (IFIS), was determined by three pathologists who were blind to tissue groups during the analysis [54].

### 4.6. Statistical Analysis

One-way ANOVA was used for normally distributed data. χ2 tests were used to assess associations of gender and lifestyle factors with the presence of adenomas at colonoscopy. Linear regression models were used to evaluate the strength of association between pairs of variables expressed by continuous numeric values. Logistic regression models were used to analyze the association between presence of adenomas at colonoscopy as dependent dichotomous variable and the other continuous variables. Multivariate regression models were created in order to evaluate the independent correlation of each variable, identified with univariate analyses, with the level of autophagy/inflammation in the colorectal mucosa of the patients examined. All statistical analyses were performed through the statistical software platform SigmaPlot© v.12 (Systat Software, Inc., San Jose, CA 95131, USA). *p* < 0.05 was considered significant.

## Figures and Tables

**Figure 1 ijms-23-05211-f001:**
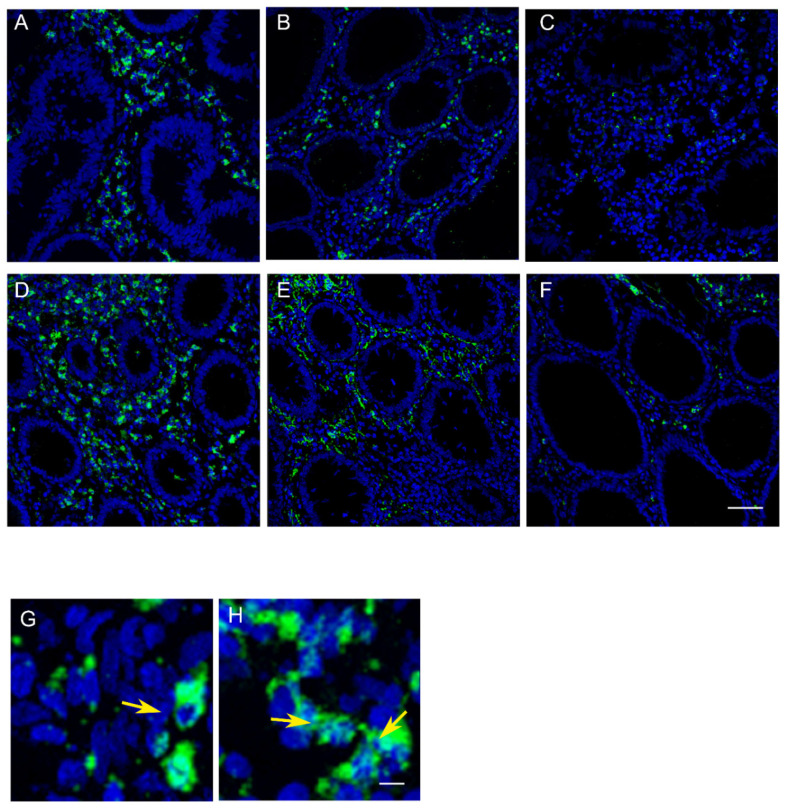
(**A**–**C**) Section of normal colorectal mucosa labeled with DAPI (blue) and anti-Myeloperoxidase (green) by confocal microscope. (**A**) Right tract of the colon: Many stromal cells distributed among the crypts show marked staining; epithelial cells are unlabeled. (**B**) Left tract of the colon: the staining clearly decreases with respect to the ascending tract. (**C**) Rectum: in this part of the bowel only a few cells are positive for myeloperoxidase labelling. (**D**–**H**) Section of normal mucosa labeled with DAPI (blue) and anti-MAPLC3 (green) by confocal microscope. (**D**) Right tract of the colon: MAPLC3 protein is localized mainly in stromal cells at the cytoplasmic level; the staining patterns varies from a diffuse mode to a granular one and is very evident (**G**). (**E**) Left tract of the colon: The immunostaining is significantly decreased and the puncta formation of MAP-LC3 in the autophagosome is clearly detectable, especially around the nucleus (**H**). (**F**) Rectum: rare cells spread in the stroma show bright tiny or large aggregates (autophagosomes). Scale bar: 100 µm for A up to F; Scale bar: 10 µm for G, H.

**Figure 2 ijms-23-05211-f002:**
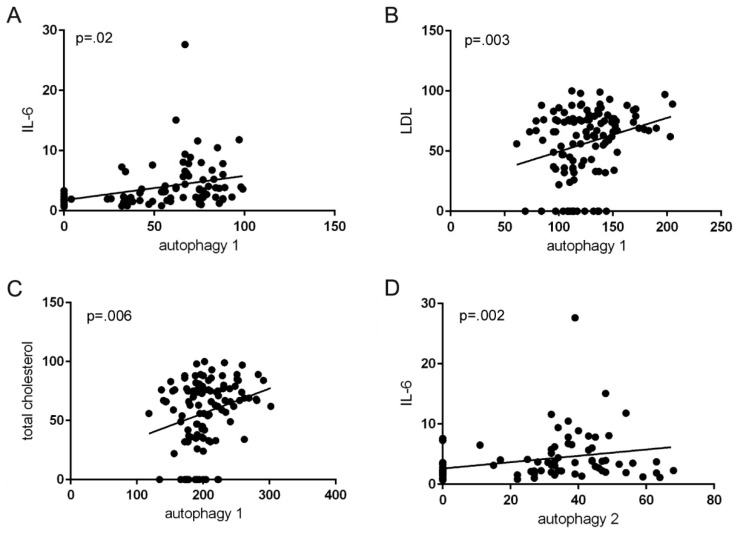
Simple scatter plots with regression line indicating significant positive direct correlations between IFIS for MAP-LC3 in the right colon (autophagy 1) and serum Il-6 (**A**), LDL (**B**), and total cholesterol (**C**), and between IFIS for MAP-LC3 in the left colon (autophagy 2) and serum IL-6 (**D**).

**Table 1 ijms-23-05211-t001:** Semiquantitative myeloperoxidase protein expression in normal colorectal mucosa of patients undergoing colonoscopy, by immunofluorescence and confocal microscopy analyses (IFIS score). NMRC, normal mucosa of the right colon; NMLC, normal mucosa of the left colon; NMR, normal mucosa of the rectum.

Groups	*n*	IFIS Mean (SD)	*p*
NMRC	116	101.5 (10.9)	<0.001
NMLC	116	69.9 (13.9)	
NMR	116	13.5 (9.7)	

**Table 2 ijms-23-05211-t002:** Semiquantitative MAPLC3 protein expression in normal colorectal mucosa of patients undergoing colonoscopy, by immunofluorescence and confocal microscopy analyses (immunofluorescence intensity score, IFIS). NMRC, normal mucosa of the right colon; NMLC, normal mucosa of the left colon; NMR, normal mucosa of the rectum.

Groups	*n*	IFIS Mean (SD)	*p*
NMRC	116	57.7 (27.9)	<0.001
NMLC	116	27.6 (19.8)	
NMR	116	2.3 (4.9)	

**Table 3 ijms-23-05211-t003:** Demographic, anthropometric, and clinical variables of the patients enrolled in the study. Abbreviations: BMI, body mass index; Chol tot, total cholesterol; CRP, c-reactive protein; DBP, diastolic blood pressure; F, female subjects; Glyc, glycaemia; HDL, high density lipoprotein cholesterol; LDL, low density lipoprotein cholesterol; M, male subjects; Std Dev, standard deviation; TRGL, triglycerides; SBP systolic blood pressure; w/h ratio, waist to hip ratio. For dichotomous variables, Y/N means presence/absence of the specific condition in the patients, namely, subjects with or without adenomas, tobacco use, alcohol consumption, and sedentary lifestyle.

Continuous Variables	Size	Missing	Mean	Std Dev	Max	Min	
Age in years	116	0	60.844	9.264	82	44	
Weight Kg	116	0	77.766	17.618	125	49	
Height in meters	116	0	1.678	0.0884	1.91	1.5	
BMI Kg (m^−2^)	116	0	27.416	4.583	38.1	18.8	
Waist cm	116	0	99.516	15.489	129	65	
Hip cm	116	0	102.547	9.791	120	81	
w/h ratio	116	0	0.968	0.0999	1.17	0.75	
SBP mmHg	116	0	146.719	18.457	180	110	
DBP mmHg	116	0	83.359	10.429	105	60	
Glyc mg/dL	116	0	93.594	18.281	194	64	
Chol tot mg/dL	116	0	196.75	19.948	243	156	
HDL mg/dL	116	1	49.873	10.723	77	32	
LDL mg/dL	116	1	120.349	16.805	154	84	
TRGL mg/dL	116	0	133.156	45.543	220	57	
CRP mg/dL	116	0	0.573	0.444	2.1	0.2	
IL-6	116	0	4.215	3.831	27.649	0.84	
**Dichotomous Variables**	**Number**		**%**		**Variables**		**M/F(%)**
Sex M/F		64/52		51.6/48.4	Presence of adenomas	37/14(72.6/27.4)	
Adenoma Y/N		51/116		43.9/56.1			
Smoke Y/N		43/73		37.5/62.5			
Alcohol Y/N		77/39		67.2/32.8			
Sedentary Y/N		49/67		42.2/57.8			
**Dichotomous Variables**	**Number**		**%**		**Variables**		**M/F(%)**
Sex M/F		64/52		51.6/48.4	Presence of adenomas	37/14(72.6/27.4)	
Adenoma Y/N		51/116		43.9/56.1			
Smoke Y/N		43/73		37.5/62.5			
Alcohol Y/N		77/39		67.2/32.8			
Sedentary Y/N		49/67		42.2/57.8			

**Table 4 ijms-23-05211-t004:** Numeric distribution of patients with and without adenomas in relation to lifestyle. None of the number of subjects per variables significantly differs between the two groups.

	Variables	Positive	Negative	*p*
**Adenoma Group**	Cigarette smoking	20	19	0.85
	Alcohol use	44	17	0.28
	Sedentary lifestyle	30	28	0.25
**No-Adenoma Group**	Cigarette smoking	24	21	0.85
	Alcohol use	43	24	0.28
	Sedentary lifestyle	27	37	0.25

**Table 5 ijms-23-05211-t005:** MAPLC3 protein expression in the normal mucosa of the right colon (NMRC) and in the left colon (NMLC) according to the presence of at least one colorectal adenoma at colonoscopy, by logistic regression.

Variable	Coefficient	Standard Error	*p*	Odds Ratio	95% Confidence Interval
**NMRC**	0.020	0.007	0.008	1.0202	1.0052–1.0354
**Constant**	−1.409	0.493	0.004		
**NMLC**	0.022	0.022	0.026	1.0225	1.0027–1.0427
**Constant**	−0.869	0.435	0.012		

**Table 6 ijms-23-05211-t006:** Epithelial Immunofluorescence Intensity score (IFIS) for MAP-LC3 in the right colonic mucosa according to the presence of at least one colorectal adenoma at colonoscopy (adenoma), serum interleukin-6 levels (IL-6), serum low-density lipoprotein cholesterol levels (LDL-Chol), and serum total cholesterol levels (Total Chol).

Variable	Coefficient	Standard Error	*p*
**Constant**	15.547	15.490	0.318
**adenoma**	10.752	5.172	0.040
**Il-6**	1.732	0.696	0.014
**LDL-Chol**	−0.074	0.213	0.730
**Total Chol**	0.197	0.171	0.253

## Data Availability

The data presented in this study are available on request from the corresponding author. The data are not publicly available due to ethical restrictions.

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
