# Peer review of "Expression of Autophagic and Inflammatory Markers in Normal Mucosa of Individuals with Colorectal Adenomas: A Cross Sectional Study among Italian Outpatients Undergoing Colonoscopy"

_ijms, 2022, doi:10.3390/ijms23095211_

Round 1

Reviewer 1 Report

The Authors attempted to explore the link between  autophagy, systemic inflammation and presence of colorectal adenomas in cases series with positive or negative colonoscopy, by using specimens collected from different sites of the digestive tract and by measuring markers in serum and tissue.

The Methods are clearly presented. The Results consist of a series of marker expression analyses. 

1. Here the Authors are asked to comment on the paradoxical pro-tumoral and anti-tumoral role of autophagy with focus on colorectal cancer and within the context of the results presented in this work, enalrging the list of references regarding this theme.

2. Please also explain the possible clinical implications of your findings, how they could be applied to the clinical routine.

Reviewer 2 Report

The authors investigate associations between metabolic markers, autophagic and inflammatory changes, and the presence of adenomas in normal mucosa samples of the colon and rectum. Using immunofluorescence microscopy and lipid profiling, the authors identify an association between serum Il-6, expression of the autophagic marker MAPLC3 and the presence of adenomas in the left side of the colon. This paper addresses the emerging area of assessing the role of inflammatory responses in cancer development. While this paper suggests novel avenues for biomarker research in CRC, it does require some further fine tuning of the manuscript.

  1. The manuscript requires editing around the use of the English language (ie. Line 43 -
    “life-threatening attitude”, Line 72 – “needs that cancer cells acquire”, Line 132 – “As regards the semiquantitative”)
  2. Line 58: there is an open bracket but no closed bracket. This needs to be fixed.
  3. Results 2.1.1: This is the first time myeloperoxidase (MPO) is mentioned in the manuscript. There is no context around why the author is investigating the expression of this protein. This needs to be addressed in the introduced so that the reader understands why this is being investigated.
  4. Results 2.1.2: The author discusses two distinct autophagic features observed in the immunofluorescence staining however, these two different features are difficult to distinguish in the images in figure 1. I would suggest that the authors provide images at a higher magnification so that these features are easily identifiable.
  5. The results section jumps from 2.1.2 to 2.1.4. There appears to be a whole section missing who this has not been updated by the authors.
  6. Table 3: the commas should be stops in the values for the Mean and Std Dev? Ie. The mean age in years read 60.844 as opposed to 60,844.
  7. Figure 2: I would recommend that the p-values are positioned with their relevant scatter plots and not in the figure legend.
  8. I see that the IFIS values were determined by three independent pathologists. Did the authors consider a more accurate method of quantifying intensity in the immunofluorescence images via image analysis software such as image j?
  9. Did the authors consider multivariate analyses looking for the different variables investigated in this paper?

Round 2

Reviewer 2 Report

I thank the authors for addressing my comments and feel that the manuscript has greatly improved.